# Forced Straightening of the Back Does Not Improve Body Shape

**DOI:** 10.3390/diagnostics14030250

**Published:** 2024-01-24

**Authors:** Wojciech Piotr Kiebzak, Sun-Young Ha, Michał Kosztołowicz, Arkadiusz Żurawski

**Affiliations:** 1Institute of Health Sciences, Collegium Medicum, Jan Kochanowski University in Kielce, 25-516 Kielce, Poland; wojciech.kiebzak@ujk.edu.pl; 2Świętokrzyskie Centre for Paediatrics, Provincial Integrated Hospital in Kielce, 25-736 Kielce, Poland; 3Institute for Basic Sciences Research, Kyungnam University, Changwon 51767, Republic of Korea; mallows205@naver.com; 4Kielce Scientific Society, 25-736 Kielce, Poland; mkosztolowicz@onet.eu

**Keywords:** slump sitting, sternum, sacrum, kyphosis, lordosis, scapular retraction, pulling the scapulae

## Abstract

Background: Sitting for a long time causes various postural problems, such as slump sitting. It has been reported that employing a corrected sitting position while lifting the sternum is effective in improving this form of posture. We investigated how a corrected sitting posture with the lifting of the sternum is different from a forced position that is applied through the command and passive positions. Materials and methods: The postural angle of 270 subjects aged 19–23 years was measured in the passive, forced, and corrected positions using a Saunders inclinometer and a Formetric 4D system. Results: As a result, the corrected position had a small range (min–max) at all angles, but the forced position and passive position had a large range (min–max). The lumbar lordosis angle in the corrected position showed positive values throughout its range (min–max), while the other groups showed negative values, which indicates the kyphotic position of the lumbar section. In addition, the percentage error in the corrected position was small, but it presented high values in the other groups. When comparing the average angles between the groups, there were substantial changes observed between the corrected position and the other groups. It was found that the corrected position with the sternum lifted, which is applied to improve slump sitting in the clinical environment, exhibited an angle that differed from that of the forced position and the passive position. Conclusions: Our results suggest that a forced position on the command “scapular retraction” does not meet the clinical assumptions of posture correction, in contrast to the corrected position with the lifting of the sternum for the improvement of slump sitting. The accurate correction of the position of the sternum and sacrum improves the position of the spine in the sagittal plane, enabling physiological values for the kyphosis and lordosis angle parameters to be obtained. This approach combines the ease of execution and precision of the effect. The fact that this method does not require complex tools to accurately correct the body encourages the implementation of this solution in clinical practice.

## 1. Introduction

Sitting is a representative activity of daily life [1], and the sitting position and its biomechanical aspects have been extensively studied in the literature. For example, in Germany, many researchers have described its standards [2]. It has been suggested that the curves present in the sitting position maintain a physiological alignment and have the same parameters as those present in the standing position [3,4]; however, some commonly accepted forms of sitting do not meet these assumptions [5,6].

Adults regularly spend 6 h a day in an inactive position [7], but this can be as much as 8–12 h [8,9]. Approximately 40% of 3-month-old infants and 90% of 24-month-old children watch TV or videos, which affects their sedentary time [10]. About 83% of adolescents also spend more than 10 h a day in a sitting position because of activities such as schoolwork, watching TV, using a computer, and playing tablet games [11,12,13]. This lifestyle can lead to reduced physical activity, poor posture, and musculoskeletal disorders [14,15]. Even though a sedentary lifestyle is a primary determinant of postural disorders, the fact that modern humans spend most of their day in a sitting position remains unchanged [16].

Regardless of one’s age and situation, a common problem caused by a sedentary lifestyle is a slanted silhouette, with the head and shoulders shifted forward, the pelvis rotated posteriorly, and the spine bent in the so-called slumped sitting posture [2,17,18]. This posture lengthens the thoracolumbar spine [19] and relaxes the trunk muscles [18]; it is accompanied by the altered posture of the spine and pelvis and changes in the position and kinematics of the scapula, thus leading to various musculoskeletal disorders [20,21,22,23,24]. A bent spine negatively affects spinal proprioception [25] and causes back pain [26]. In addition, changes in hyoid muscle tone [27], difficulty swallowing food [28], and bowel problems [29] have also been reported.

The optimal sitting posture is associated with a forward pelvic tilt, lumbar lordosis, and proper thorax positioning [18]. In this paper, we refer to this position as corrected. Proper pelvic positioning and lumbar lordosis are achieved more easily when, in a sitting position, the hip is set higher than the knee and the feet are thoroughly rested on the ground. To prepare the optimal sitting position, the positioning of the lower limbs is also taken into account. To achieve this position, the physiological curve of the spine should be considered; this should be close to the normal state in order to reduce energy expenditure [30]. In order to correct slumped sitting in clinical practice, scapular retraction reinforcement is applied to straighten the curved back, and commands such as “straighten the back” are often given. However, these methods may lead to incorrect posture because they do not consider the overall alignment of the spine. In addition, the command for posture correction is inappropriate because the guidance given to the subjects is not clear, and whether the subject has adopted the optimal position of the spine is unknown [31,32]. These inappropriate methods lead to an incorrect sitting posture in which the thoracolumbar spine is extended and the scapula is slightly retracted [18], that is, a sitting posture with bilateral scapular retraction [18,33]. This position is called forced. Therefore, the use of inappropriate methods to improve the slumped sitting posture may, rather, lead to incorrect posture, especially if they are performed at a high intensity and frequency. In order to ameliorate slumped sitting, the activation of the deep trunk muscles and the proper alignment of the entire spine in the sagittal plane should be considered [34]. However, most clinicians consider only the pelvic position [33]. Among the factors related to the spine, the sternum is related to thoracic kyphosis, and the inclination of the sacrum and pelvis is related to lumbar lordosis [35,36]. Kiebzak et al., (2022) observed fluctuations in the angle of the sternum and sacrum and reported that lifting the sternum is effective in postural correction [32]. Therefore, if we consider not only the pelvic position but also the angle of the sternum, which is located at the front of the body, when aiming to improve slumped sitting [37], we can efficiently deal with the overall alignment of the spine. Therefore, this investigation aims to determine how a corrective posture in which the sternum is lifted for the optimal alignment of the sitting posture is different from the forced position and passive position, in which commands are applied. 

## 2. Materials and Methods

### 2.1. Subjects

This study comprised 270 subjects. The age of the subjects varied from 19 to 23 years. The average BMI of the study population was 21.72 (±1.36). This study was performed without regard to sex because previous reports showed that there were no differences in the assessed parameters among females and males [32]. The inclusion criteria were as follows: those in good health, without back pain, and with a normal chest and spinal structure. The exclusion criteria were as follows: those whose sacral joint position was difficult to determine, those with a neuromuscular disease, those who had received specialized treatment for body posture disorders, those who had been diagnosed with scoliosis, those who had previously undergone spinal surgery, and those who were taking pain medication.

The participants of the experiment were presented with the study protocol, the purpose of the study was discussed, the security rules and privacy policy were introduced, the possibility of using photographs was discussed, and the research methodology was presented. All the subjects gave their voluntary, conscious, and written consent to participate in the experiment. This study was approved by the Bioethical Committee of the Faculty of Wydział Lekarski i Nauk o Zdrowiu Uniwersytetu Jana Kochanowskiego (Faculty of Medicine and Health Sciences of the Jan Kochanowski University) in Kielce No. 17/2016.

### 2.2. Methods

The researchers had no involvement in the selection of the study participants, and the selection of participants was random. During the physical inspection, the subject sat in a chair with an even load on the ischial tuberosities, with the hip joint slightly higher than the knee joint. The feet were placed evenly on the ground, with the hips separated. The upper limbs were placed loosely with the hands lying on the thighs. To select the most favorable position according to Mork et al., (2009) [38], the subjects assumed each position three times.

In the selection of the research tools, the safety of the participants (particularly important because of the young age of the respondents), the accuracy of the measurements, a short exposure time, and the ability of other researchers to reproduce the study were important criteria.

Measurements were taken in a sitting position in the following setting:a.In a corrected, active position, without a backrest (Figure 1). The corrected position was assumed in accordance with the examiner’s instructions. Attention was paid to the complete, active, and physiological alignment of the spine in a way that required the subject to exert the least amount of effort. The activities performed included lifting the sternum, increasing the anterior pelvic tilt, positioning the head at the axis of the spine, setting the shoulder blades in a physiological position, and slightly inclining the torso [37].

b.In a forced position, where the instructions were as follows: “Sit up straight, intensively pull back your shoulder blades” (Figure 2).

c.In a passive, free position without a backrest or muscle involvement and with posterior pelvic tilt; this is the so-called passive position (Figure 3).

Ⅰ and II were measured when the subject assumed the set position.

I.The angular parameters of the position of the sternum body (called the α angle) and the angular parameters of the sacrum position (called the β angle) in relation to the horizontal line, constituting the sagittal axis of the body. Saunders digital inclinometer (Baseline Digital Inclinometer Range of Movement Measurement Tool, New York, NY, USA) was used to conduct the assessment. The measuring tool that was used is characterized by a high measurement accuracy. The measurement resolution is 0.1 degrees, and the measurement accuracy is ±1 degree;II.The angular parameters of the thoracic spine curves (kyphosis angle called ω_1_) and lumbar spine curves (lordosis angle called ω_2_) measured using the DIERS Formetric 4D system (DICAM 3) (Figure 4). The resolution of the device is 0.01 degrees, and the accuracy is 0.25 degrees [39]. The DIERS system uses raster stereography, so it is free from any radiation. The DIERS Formetric is a light optical visualizing system based on video raster stereography. Therefore, the system comprises a light projector that creates a line grid on the back of the patient, which is noted with an imaging unit. Computer software evaluates the line bend and creates a three-dimensional model of the surface that is analogous to a plaster cast using the method of photogrammetry.

To determine the value of angle α (position of the sternum body), the feet of a Saunders inclinometer were placed on the front surface of the sternum body. To determine the value of the β angle (sacral position), one foot of the Saunders inclinometer was placed on the upper edge of the sacrum, and the other foot was placed on the surface of the medial sacral crest (Figure 5).

Regarding the size of angle α, the intraobserver repeatability (the difference in measurements performed ten times by the same researcher) SEM was 2.4°. In terms of the intraobserver repeatability, the measurement of the sacrum angle had a good SEM result of 3.7°. The intraclass correlation coefficient (ICC—measurements performed by three researchers) for the measurement of the sternal body inclination angle was 0.86. The chi-square test showed no significant differences between individual researchers; CI: 0.74/0.91. The interobserver repeatability coefficient of the ICC was 0.90 for the measurement of the sacral inclination angle when assessed by three investigators. In this case, the chi-square test also showed no significant differences between individual researchers; CI: 0.79/0.93. The reliability of Cronbach’s alpha test measurements was high; this amounted to 0.86 for the measurement of the sternal shaft angle and 0.91 for the sacrum angle [32]. As the above values show, although the inclinometer is a simple tool and subject to errors resulting from the human factor, it can be precise. It was used in this work to use the developed assumptions in small institutions that do not have specialized equipment. On this basis, common sense was calculated:(a)Of sacral angle and sternal angle: γ = β − α;(b)Of sternal angle and thoracic kyphosis: γ_1_ = 180 − (α + ω_1_);(c)Of sternal angle and lumbar lordosis: γ_2_ = 180 − (β + ω_2_).

### 2.3. Statistical Analysis

The results obtained during this study were statistically analyzed with the Statistica 13.3 StatSoft software. The level of statistical significance for the performed examination was assumed to be *p* < 0.05. The basic descriptive statistics were calculated. The normality of the distribution was assessed using the Shapiro–Wilk test. A one-way ANOVA test was used to compare the differences in the dependent variables between groups. A post hoc Tukey’s test was used to detect differences in the relative angles between specific groups. Pearson’s correlation analysis was used to examine the relationship between the position and angle. The confidence intervals of common sense were established [32].

## 3. Results

### 3.1. Measured Angles Based on 3 Positions

The angles measured in each position are shown in Table 1.

### 3.2. Percentage Errors

In common sense, according to the descriptive statistics presented in Table 1, the percentage errors (E_p_%) showed that the corrected position had fewer errors than the other positions. The γ of the corrected position showed a lower error than the γ of the other positions (Table 2).

### 3.3. Comparison of the Angle between Groups

For α, ω_1_, and ϒ_2_, the passive position had the largest angle, and the corrected position had the smallest angle (*p* < 0.05). For β, the corrected position had a larger angle than the passive and forced positions (*p* < 0.05). For ω_2_, ϒ, and ϒ_1_, the corrected position had the largest angle, and the passive position had the smallest angle (*p* < 0.05) (Table 3).

### 3.4. Correlation between Angles

In the corrected position, ϒ showed a positive correlation with the sacral angle (r = 0.892, *p* < 0.05), and ϒ_2_ showed a negative correlation with lumbar lordosis (r = −0.811, *p* < 0.05).

In the forced position, ϒ showed a negative correlation with the sternal angle and a positive correlation with the sacral angle (r = −0.843, *p* < 0.05). ϒ_1_ showed a negative correlation with the sternal angle (r = −0.890, *p* < 0.05) and kyphosis (r = −0.930, *p* < 0.05). ϒ_2_ showed a negative correlation with lordosis (r = −0.705, *p* < 0.05).

In the passive position, ϒ showed a negative correlation with the sternal angle (r = −0.812, *p* < 0.05) and a positive correlation with the sacrum (r = 0.801, *p* < 0.05). ϒ_1_ showed a negative correlation with the sternal angle (r = −0.763, *p* < 0.05) and kyphosis (r = 0.844, *p* < 0.05). ϒ_2_ was negatively correlated with lordosis (r = −0.751, *p* < 0.05) (Table 4).

### 3.5. Confidence Intervals

The confidence intervals that were calculated when measuring the angles according to the postures are shown. The corrected position had a smaller confidence interval than the other positions (Table 5).

## 4. Discussion

Because of the increase in smartphone and computer use and students’ studies, activities that involve sitting for a long time cause the development of an incorrect posture, such as a slumped posture. Various interventions are currently being applied to correct this, but inappropriate methods may cause an incorrect posture. Although the analysis of slumped posture has been reported in previous studies [40], an efficient solution that is able to improve posture has not been suggested. Therefore, we have attempted to suggest the optimum posture by measuring the angles of the sternum, sacrum, kyphosis, and lordosis in various postures to ameliorate the slumped posture.

One of the fundamental characteristics determining the quality of body posture is the curvature of the spine in the sagittal plane [41]. If the angle in the sagittal plane deviates from its normal state because of poor posture when sitting or standing for a long time, spinal diseases can occur [42]. In body alignment, the curve of the spine in a sitting position should resemble that of an “ideal” standing position [40]. In this study, the corrected posture includes the correct body alignment, and the forced and passive postures are misaligned. The corrected position is a posture in which the sternum is lifted and the pelvis is tilted anteriorly, and the usefulness of this posture has been proven in previous studies [32]. In normal alignment, the kyphotic angle of the thoracic spine in the sagittal plane is about 45° [43,44], but this is about 20–40° in teenagers [45,46]. Various interventions have been applied to reduce the thoracic kyphosis that is caused by slump sitting. However, if repetitive motions are applied without considering correct alignment, the kyphosis angle of the spine decreases, causing hypokyphosis [47] or lumbar kyphosis. According to the results of this study, the average value of the kyphosis angle in the corrected position was similar to that in the forced position, but the passive position showed a larger kyphosis angle. The corrected position range (min–max) was 26.5–49.7°, but the forced position range (min–max) was 12.3–70.9° (Table 1). Clément et al. (2013) reported that the thoracic kyphosis angle of healthy adolescents ranges from 27.9 to 44.2°, which is similar to the results obtained in this study regarding the corrected position [48]. The range of the kyphotic angle in the forced position was greater than that in the corrected position. Czaprowski et al., (2014) reported that when posture correction was performed on command, the neutral position of the spine could not be adopted because the lumbar and thoracic vertebrae were extended, and the kyphosis of the lower thoracic spine was rather reduced [31]. The kyphosis shape of the lower thorax plays a significant role in maintaining the rotational stability of the spine [49]. In this study, we found that the range of kyphosis in the forced position was wide because the correct spine position could not be adopted when the command was presented for the forced position.

In lumbar lordosis, the average of the corrected position was 37.7°, the forced position was 12.6°, and the passive position was −5.3°. The corrected position range (min–max) was 28.3–46.6°, but the forced and passive positions had rather negative values, indicating the occurrence of lumbar kyphosis. O’Sullivan et al., (2006) reported kyphosis of about 20° in the thoracic spine and lordosis of about −25° in the lumbar spine during lumbopelvic sitting [18]. The forced position that is initiated by verbal command is similar to the thoracic sitting proposed by O’Sullivan et al. (2006). This posture applies a high-pressure load to the spine as a result of the action of the global spinal muscle, and the efficiency ratio is small. In addition, this posture is associated with the increased activity of the thoracic erector muscles and iliac costal muscles at the thoracic 4 level. This may result in greater stress being caused to joint and ligamentous structures, greater compressive loads on the cervical–thoracic spine, and a higher risk of potential discomfort [18,32,40]. Therefore, it is thought that improper posture correction can result in the kyphosis of the lumbar spine, cause an overload of the lumbar spine, and lead to the need for radiculopathy.

The sternum angle affects thoracic kyphosis, and the sacrum angle affects lumbar lordosis [48,50]. In particular, a correlation between the lumbar spine and sacropelvic alignment has been demonstrated in a variety of populations comprising healthy children and adults and patients with scoliosis [51,52,53]. Kiebzak et al., (2022) suggested that monitoring the simultaneous movement of the thoracic and lumbar spine, as well as the sternum and sacrum, is useful for understanding movement [32]. O’Sullivan et al., (2006) said that the pelvic angle is important with regard to the activity of the antigravity muscles [18]. In this study, the angles measured in the corrected position significantly differed from the angles measured in the forced and passive positions (Table 3). In addition, the corrected position, forced position, and passive position exhibited a significant correlation with the common sense and angle (Table 4). In all postures, ϒ was significantly correlated with the sacrum angle and ϒ_2_ with lumbar lordosis. In the forced and passive positions, ϒ_1_ was significantly correlated with kyphosis; as the kyphosis angle increases, ϒ_1_ decreases. As shown in Table 1, these results indicate that the range of the kyphosis angle in the forced position and passive position is larger than that in the corrected position. The confidence intervals at three positions show a narrower range for the corrected position than for the forced and passive positions, which means that the homogeneity of the measurements is greater. In addition, the confidence intervals for thoracic kyphosis and lumbar lordosis in the corrected position are close to the reference values that were reported in the literature, thus proving the clinical value of this position (Table 5) [54,55].

Taken together, there was a significant difference in the sagittal angle of the corrected position compared to those of the forced and passive positions. With regard to ameliorating the slump position in clinical practice, the passive position and the position forced via command were not effective in aligning the sagittal plane. Therefore, we suggest that the sternum oblique is lifted for the corrected position and that the correct alignment is maintained through pelvic anterior tilt and the setting of the shoulder blades. The assumptions presented in this paper apply not only to young adults but also to the entire population. Earlier studies [32] indicate that they can be effectively applied in the pediatric population because the difference in the angular values of the sternum and the sacrum in children is less than 2°. From a practical point of view, it is significant that more than 86% of people indicate that the corrected position is comfortable and easy to adopt and maintain [37].

Interdependent movements in the sternum and the sacrum cause changes in the angles of thoracic kyphosis and lumbar lordosis; this is a relationship that has been described in detail using Euclidean geometry [56]. Moreover, these changes require precise monitoring during the treatment process, as they determine changes in other elements of the musculoskeletal system [57]. This fact should be a significant element in the clinical observation of body posture while sitting. The interestingness of the concept also results from the fact that the correction of the spine position in the sagittal plane also generates a correction of the spine position in the frontal and transverse planes [58], which should encourage other authors to verify this position. The implementation of a sternal body angle of about 64° in relation to the sagittal axis of the body, as one of the goals of postural correction, may be key to the correction of body posture while sitting in clinical practice.

The limitations of this study are that it only assessed three sitting positions and that other variables able to evaluate posture other than the angle were not measured. In future studies, supplementary studies should be conducted. This study did not compare the parameters of the spine position in the sitting position with the same parameters in the standing position; this could provide valuable information, as much more research is carried out in the standing position. The narrow age range of the respondents is worth noting, as this was dictated by the need to standardize the research group; the implementation of the described solutions in other age groups and various clinical dysfunctions requires verification.

## 5. Conclusions

Our results suggest that the forced position on command, namely, “scapular retraction”, does not meet the clinical assumptions of posture correction; this is in contrast to the position that was corrected by lifting the sternum to improve slump sitting. The accurate correction of the position of the sternum and sacrum improves the position of the spine in the sagittal plane, allowing physiological values of the kyphosis and lordosis angle parameters to be obtained. This approach combines the ease of execution and precision of the effect. The fact that this method does not require complex tools to accurately correct the body encourages the implementation of this solution in clinical practice.

## Figures and Tables

**Figure 1 diagnostics-14-00250-f001:**
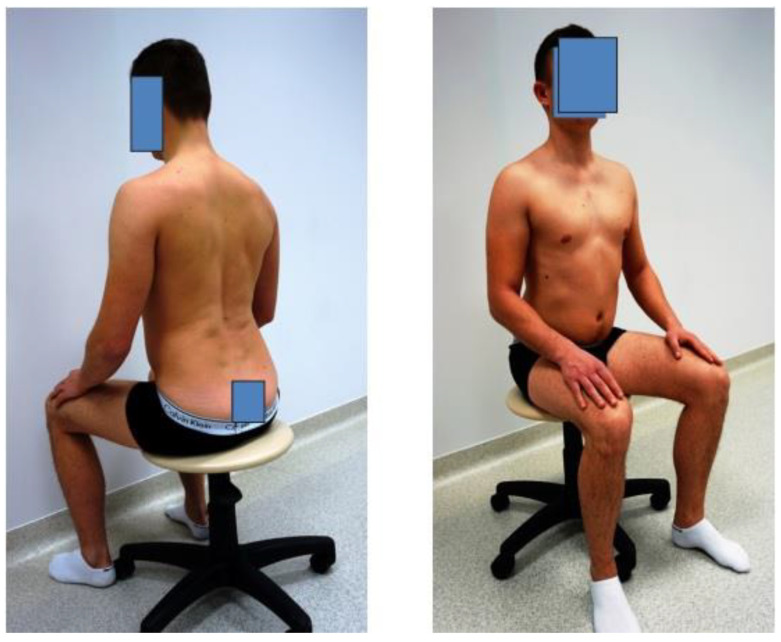
Corrected position.

**Figure 2 diagnostics-14-00250-f002:**
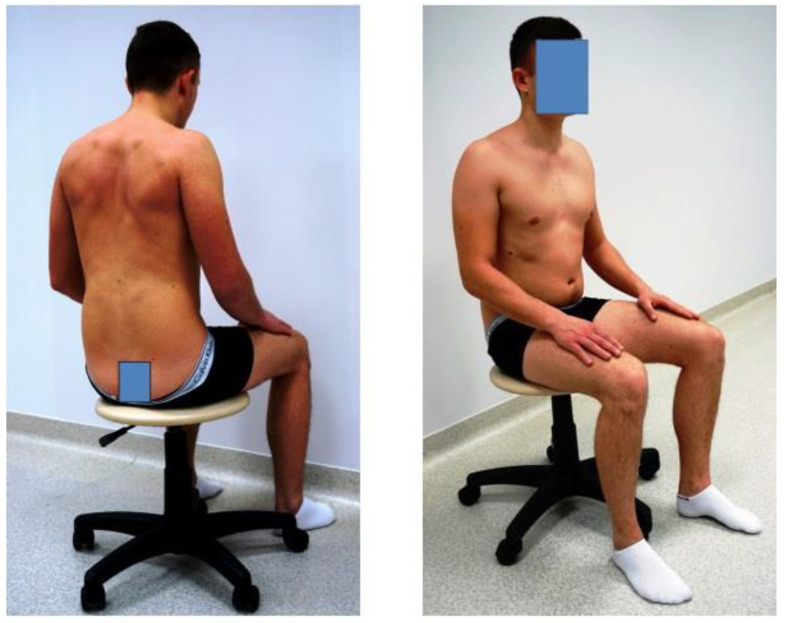
Forced position.

**Figure 3 diagnostics-14-00250-f003:**
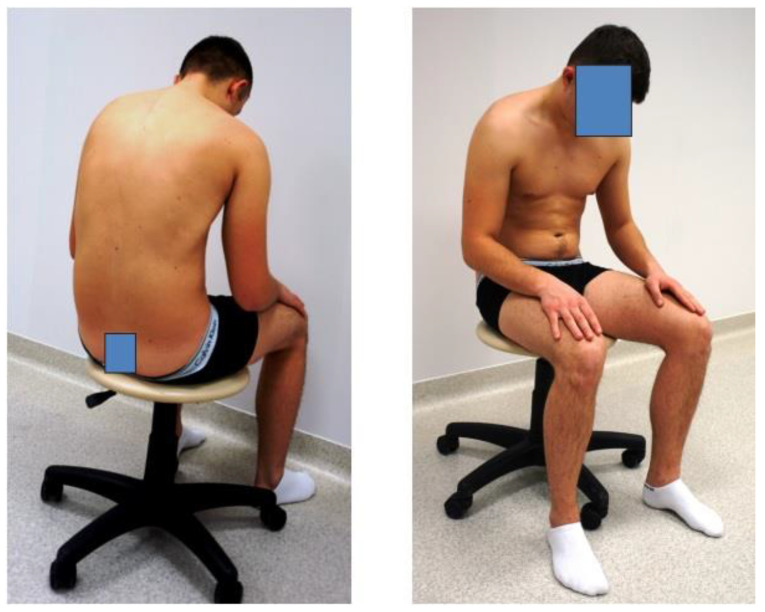
Passive position.

**Figure 4 diagnostics-14-00250-f004:**
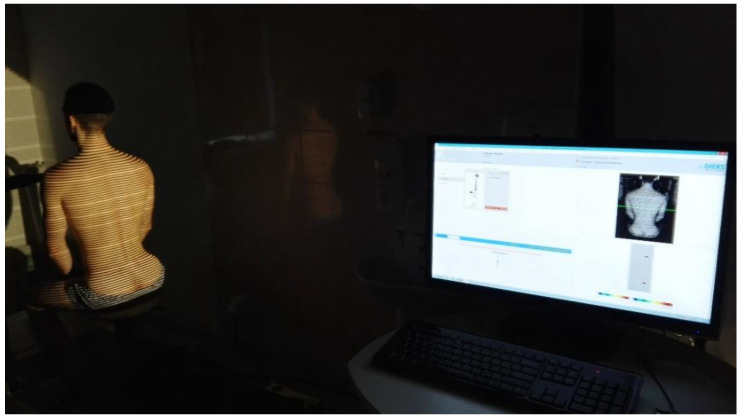
Measurements with the DIERS Formetric system.

**Figure 5 diagnostics-14-00250-f005:**
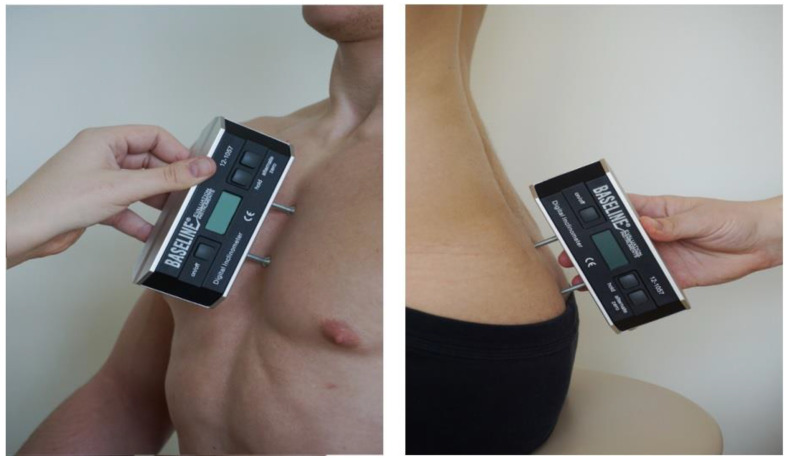
Measurement of the angle of the sternum and the angle of the sacrum [32].

**Table 1 diagnostics-14-00250-t001:** Angles measured in each of the 3 positions (°).

	Corrected Position					Forced Position					Passive Position				
	M (SD)	Median	Mode	Min	Max	M (SD)	Median	Mode	Min	Max	M (SD)	Median	Mode	Min	Max
α	64.20 (2.44)	64.00	65.00	60.00	69.70	70.52 (11.05)	74.10	74.90	47.30	88.30	84.19 (11.02)	83.20	91.00	58.20	111.60
β	113.39 (4.98)	113.40	1150	100.0	127.0	91.22 (8.83)	89.70	87.40	74.10	125.50	81.64 (10.82)	80.20	72.00	62.00	118.80
ω_1_	41.39 (4.33)	41.73	41.50	26.46	49.70	43.41 (13.74)	47.00	55.20	12.30	70.90	57.57 (13.34)	59.00	63.30	20.50	89.00
ω_2_	37.68 (3.51)	37.68	38.60	28.30	46.60	12.77 (12.31)	11.40	4.30	−15.80	39.90	−5.14 (15.87)	−6.90	−27.30	−39.25	38.40
ϒ	49.19 (5.38)	49.00	48.00	34.60	61.30	20.70 (15.83)	17.70	−5.40	−9.40	65.20	−2.54 (17.64)	−3.30	1.00	−46.70	44.00
ϒ_1_	74.41 (4.84)	74.50	73.40	62.50	87.40	66.06 (22.60)	63.50	51.00	27.30	114.80	38.24 (19.78)	39.50	15.40	−10.50	93.20
ϒ_2_	78.11 (4.14)	78.10	77.60	66.00	88.20	96.70 (15.40)	96.60	112.30	62.90	136.70	100.96 (15.59)	101.90	108.8 0	61.80	139.40

M—mean; SD—standard deviation; α—sternal angle; β—sacrum angle; ω_1_—thoracic kyphosis; ω_2_—lumbar lordosis; ϒ—sacral angle − sternal angle; ϒ_1_—180 − (sternal angle + thoracic kyphosis); ϒ_2_—180 − (sacral angle + lumbar lordosis).

**Table 2 diagnostics-14-00250-t002:** Comparison of percent error between empirical median and theoretical median in sitting positions.

	Corrected Position			Forced Position			Passive Position		
	M_e,t_ (°)	M_e,emp_ (°)	E_p_% (%)	M_e,t_ (°)	M_e,emp_ (°)	E_p_% (%)	M_e,t_ (°)	M_e,emp_ (°)	E_p_% (%)
ϒ	49.19	49.00	0.40	20.70	17.70	16.38	−2.54	−3.30	22.22
ϒ_1_	74.41	74.50	0.26	66.06	63.50	4.92	38.24	39.50	3.40
ϒ_2_	78.11	78.10	0.12	96.70	96.60	0.31	100.96	101.90	0.99

Me,t—theoretical median; Me,emp—empirical median; Ep%—percentage errors; ϒ—sacral angle—sternal angle; ϒ_1_—180 − (sternal angle + thoracic kyphosis); ϒ_2_—180 − (sacral angle + lumbar lordosis).

**Table 3 diagnostics-14-00250-t003:** Comparison of angles between groups (°).

Angle	Corrected Position ^a^	Forced Position ^b^	Passive Position ^c^	F	*p*	Post Hoc
α	64.20 ± 2.44	70.52 ± 11.05	84.19 ± 11.02	334.752	0.000	a < b < c
β	113.39± 4.98	91.22 ± 8.83	81.64 ± 10.82	161.445	0.000	a > b = c
ω_1_	41.39 ± 4.33	43.41 ± 13.74	57.57 ± 13.34	965.185	0.000	a < b < c
ω_2_	37.65 ± 3.51	12.77 ± 12.31	−5.14 ± 15.87	890.653	0.000	a > b > c
ϒ	49.19 ± 5.38	20.70 ± 15.83	−2.54 ± 17.64	909.821	0.000	a > b > c
ϒ_1_	74.41 ± 4.84	66.06 ± 22.60	38.24 ± 19.78	310.125	0.000	a > b > c
ϒ_2_	78.11 ± 4.14	96.70 ± 15.40	100.96 ± 15.59	237.499	0.000	a < b < c

*p* < 0.05. Values are presented as mean ± standard deviation. α—sternal angle; β—sacrum angle; ω_1_—thoracic kyphosis; ω_2_—lumbar lordosis; ϒ—sacral angle—sternal angle; ϒ_1_—180 − (sternal angle + thoracic kyphosis); ϒ_2_—180 − (sacral angle + lumbar lordosis).

**Table 4 diagnostics-14-00250-t004:** Correlation between common sense and angles according to position.

Common Sense	Angle	Corrected Position	Forced Position	Passive Position
ϒ	α	−0.391 *	−0.843 *	−0.812 *
	β	0.892 *	0.737 *	0.801 *
ϒ_1_	α	−0.130 *	−0.890 *	−0.763 *
	ω_1_	−0.134 *	−0.930 *	−0.844 *
ϒ_2_	β	−0.210 *	−0.411 *	−0.441 *
	ω_2_	−0.811 *	−0.705 *	−0.751 *

α—sternal angle; β—sacrum angle; ω_1_—thoracic kyphosis; ω_2_—lumbar lordosis; ϒ—sacral angle—sternal angle; ϒ_1_—180 − (sternal angle + thoracic kyphosis); ϒ_2_—180 − (sacral angle + lumbar lordosis); * means a statistically significant correlation at the *p* < 0.05 level.

**Table 5 diagnostics-14-00250-t005:** 90% CI for angles measured in 3 positions.

	90% CI in the Corrected Position	90% CI in the Forced Position	90% CI in the Passive Position
α	61.00–67.90	52.8–83.25	70.95–99.05
β	107.10–119.70	81.12–102.60	68.70–96.20
ω_1_	35.30–46.82	26.10–58.56	39.09–73.16
ω_2_	32.67–42.02	−4.32–29.92	−24.52–17.30
ϒ	42.15–56.55	0.68–41.88	−26.00–20.65
ϒ_1_	68.04–81.06	38.52–98.60	12.35–65.17
ϒ_2_	72.57–83.99	76.42–116.55	79.95–120.66

α—sternal angle; β—sacrum angle; ω_1_—thoracic kyphosis; ω_2_—lumbar lordosis; ϒ—sacral angle—ternal angle; ϒ_1_—180 − (sternal angle + thoracic kyphosis); ϒ_2_—180 − (sacral angle + lumbar lordosis).

## Data Availability

Because of the confidentiality of information regarding the respondents, data are available from the corresponding author upon request.

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
