# Peer review of "Forced Straightening of the Back Does Not Improve Body Shape"

_diagnostics, 2024, doi:10.3390/diagnostics14030250_

Round 1
Reviewer 1 Report
Comments and Suggestions for Authors
Dear,
please find my comments attached.
Kind regards

Author Response
Dear reviewer,
Thank you very much for your time and valuable comments. We have implemented all your suggestions, thanks to which our manuscript is of better quality.
I will address each of your suggestions in turn:
1) In the Introduction section, specify that the ideal sitting position involves the hips positioned slightly higher than the knees, with feet consistently flat against the support surface;
Answer: Thank you for your comment. Of course, it is impossible to properly position the pelvis if the hips are not above the knees. We have updated the Methods section with this information.
2) In subsection 2.2 Methods, emphasize that non-invasive and radiation-free methodologies, inclinometer and Video-Raster-Stereography, were employed in the research, considering the young age of the participants (19-23 years);
Answer: Thank you for this suggestion. In the Methods section, we emphasized the importance of using non-invasive methods in the diagnostic process.
3) In the same subsection 2.2 Methods, in the description of sitting positions during testing, the sequence used was Figure 1 Passive position, Figure 2 Forced position, and Figure 3 Corrected position. This order does not align with the sequence presented in the subsequent tables. It is necessary to standardize the order of presentation.
Answer: Thank you very much for drawing attention to this fact. Before being submitted, the text was rebuilt several times and the disorder remained. The order of figures has been corrected to match the order of the examined items presented in the text.
Reviewer 2 Report
Comments and Suggestions for Authors
Dear corresponding Author, thanks for submitting your paper. Your research topic is very interesting because it makes light ona very practical issue of everyday life.
Some considerations in the following lines:
Abstract:
Line 25: "kyphosis appeared in the lumbar" please, what do you mean?
Conclusions: avoid the bullet point.
Introduction: Congratulations, the intro is well-written. I have just one question about it, in lines 66-67 you wrote "The optimal sitting posture is associated with anterior pelvic tilt, lumbar lordosis, and relaxation of the thorax [18].", then in line 90 "the corrective posture in which the sternum was lifted for optimal sitting". I understand that when you write about to lift up the sternum you mean starting form the slump sitting (therefore you arrive to a correct positionof the thorax). At the same time if you use your muscle to lift up the sternum you don't have a relaxed thorax. It is contraddictory. Please adjust this aspect.
Discussion:
in lines 230-231 you wrote "In body alignment, the spine curve in a sitting position should resemble that of an “ideal” standing position [40]." at the same time you did not measure the spine curves in standing position to make a comparison with the sitting values. I thin that this aspect could be very interesting. I think that it must be add in the "Limitations" section, suggesting that future studies can be performed with this aim.
Line 232 "includes optimal body alignment" regarding the previous comment the word "optimal" is not certain. Please modify.
Please expand the limitations section.
Conclusion: avoid the use of bullet point and add the practical applications of your results.
Pay attention to double dot at the end of the sentence in some parts of the paper
Author Response
Dear reviewer,
Thank you very much for your time and valuable comments. We implemented all your suggestions, which improved the quality of the manuscript. I will address each of your comments in turn:
1.Abstract; "kyphosis appeared in the lumbar" please, what do you mean?
Reply: Thank you for this comment. We have rebuilt this sentence and it now reads as follows: "The lumbar lordosis angle in the corrected position, showed positive values in the range (min-max), while the other groups showed negative values, which indicates kyphotic position of the lumbar section. "
2. Abstract; Conclusions: avoid the bullet point.
Answer: Thank you for this suggestion. The conclusions section has been rebuilt in both the abstract and main text. The conclusions are currently described in one paragraph.
3. Introduction: Congratulations, the intro is well-written. I have just one question about it, in lines 66-67 you wrote "The optimal sitting posture is associated with anterior pelvic tilt, lumbar lordosis, and relaxation of the thorax [18].", then in line 90 "the corrective posture in which the sternum was lifted for optimal sitting". I understand that when you write about to lift up the sternum you mean starting form the slump sitting (therefore you arrive to a correct positionof the thorax). At the same time if you use your muscle to lift up the sternum you don't have a relaxed thorax. It is contraddictory. Please adjust this aspect.
Answer: Thank you for your attention. You're right, this statement sounded illogical. The idea was not to use much force for correction. We have adjusted the terms throughout the text to make them sound consistent.
4. in lines 230-231 you wrote "In body alignment, the spine curve in a sitting position should resemble that of an “ideal” standing position [40]." at the same time you did not measure the spine curves in standing position to make a comparison with the sitting values. I thin that this aspect could be very interesting. I think that it must be add in the "Limitations" section, suggesting that future studies can be performed with this aim.
Answer: Thank you very much for pointing this out. You are right, we examined the curvature of the spine in a standing position and we refer to these parameters in the submitted work. However, we have never compared these parameters in different positions. As you suggested, I added the Limitations section.
5. Line 232 "includes optimal body alignment" regarding the previous comment the word "optimal" is not certain. Please modify.
Answer: Thank you for this suggestion. This has been modified and we have limited the use of such wording throughout the text.
6. Please expand the limitations section.
Answer: Thank you for your suggestion. The Limitation section has been significantly expanded. It has also been updated with your earlier suggestion.
7. Conclusion: avoid the use of bullet point and add the practical applications of your results.
Answer: Thank you for this suggestion. The conclusions section has been rebuilt in both the abstract and main text. The conclusions are currently described in one paragraph.
8. Pay attention to double dot at the end of the sentence in some parts of the paper
Answer: Thank you for this comment. We have corrected punctuation throughout the text. It has also been better organized.